# DENSE VIDEO OBJECT CAPTIONING FROM DISJOINT SUPERVISION

**Xingyi Zhou**[*]  **Anurag Arnab**[*]  **Chen Sun**  **Cordelia Schmid**
Google DeepMind

## ABSTRACT

We propose a new task and model for *dense video object captioning* – detecting, tracking and captioning trajectories of objects in a video. This task unifies spatial and temporal localization in video, whilst also requiring fine-grained visual understanding that is best described by natural language. We propose a unified model, and demonstrate how our end-to-end approach is more accurate and temporally coherent than a multi-stage pipeline combining state-of-the-art detection, tracking, and captioning models. Moreover, we propose a training strategy based on a mixture of disjoint tasks, which allows us to leverage diverse, large-scale datasets which supervise different parts of our model. Although each pretraining task only provides weak supervision, they are complementary and, when combined, result in noteworthy zero-shot ability and serve as strong initialization for additional finetuning to further improve accuracy. We carefully design new metrics capturing all components of our task, and show how we can repurpose existing video grounding datasets (*e.g.* VidSTG and VLN) for our new task. We show that our model improves upon a number of strong baselines for this new task. Furthermore, we can apply our model to the task of spatial grounding, outperforming prior state-of-the-art on VidSTG and VLN, without explicitly training for it. Our code is available at https://github.com/google-research/scenic.

## 1 INTRODUCTION

Powered by gigantic datasets and models, *language* is becoming the output modality of the most capable artificial intelligence models (Team et al., 2023; Alayrac et al., 2022; Ouyang et al., 2022; Li et al., 2023; Liu et al., 2023; Tong et al., 2024; Li et al., 2024a). Language unifies different tasks with the same output space (Raffel et al., 2020; Chen et al., 2023a), is more descriptive than discrete class labels (Wu et al., 2022a; Long et al., 2023), and naturally facilitates zero-shot prediction of novel tasks (Radford et al., 2021; Brown et al., 2020). Inspired by advances in natural language understanding, the vision community has explored language in a number of tasks including image captioning (Chen et al., 2015), dense image captioning (Krishna et al., 2017b), question answering (Antol et al., 2015), video captioning (Monfort et al., 2021) and representation learning (Radford et al., 2021). However, likely due to the scarcity of large-scale, aligned training data, we are not aware of any existing single vision-language model that unifies both fine-grained **spatial**- (by detecting objects) and **temporal**- (by reasoning across time in videos) understanding.

In this paper, we propose a new task and model for *dense video object captioning* (Dense VOC) – the task of generating captions of trajectories of all objects from video (Fig. 1). Dense VOC requires understanding across space, time, and language (Fig. 2), and is therefore a superset of existing vision tasks, namely object detection (Everingham et al., 2015; Lin et al., 2014), multi-object tracking (Dendorfer et al., 2021; Dave et al., 2020) and captioning (Chen et al., 2015). It offers a broad range of applications, such as sports analysis, wildlife monitoring, and behavioral analysis.

A prominent challenge for training our model is that datasets with captioned trajectories are scarce. However, annotations for each sub-task, or even each combination of the sub-tasks, are abundant. For example, we can train our object proposal component using image-level object detection labels from COCO (Lin et al., 2014), and the captioning component from video-level captioning datasets like SMiT (Monfort et al., 2021). These disjoint training tasks are complementary, and in combination supervise our entire model. This enables us to perform our Dense VOC task in a zero-shot manner, and we show that we can achieve noteworthy performance despite not having access to any full,

---

[*]Equal contribution. {zhouxy, aarnab}@google.com

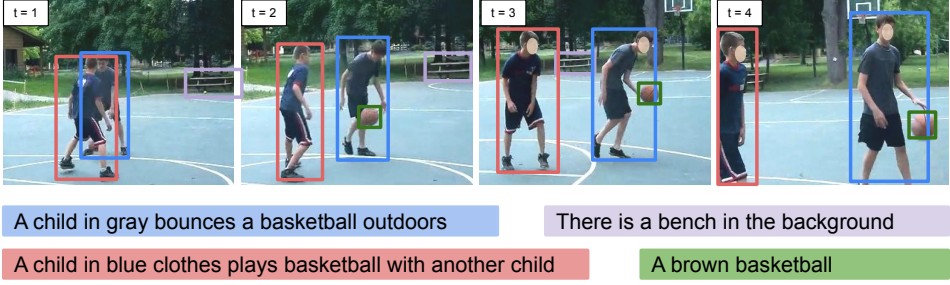

Figure 1: **Overview of the dense video object captioning (Dense VOC) task.** Given a video, we predict object trajectories (identities denoted by colors) and their natural language description. We show a video from the VidSTG (Zhang et al., 2020) validation set.

captioned object trajectories during training. Furthermore, this pretraining serves as a powerful initialization for finetuning on the full Dense VOC task, where limited annotations are available.

Another challenge in our task is to produce holistic and consistent captions for objects across frames. Note that a baseline of applying a strong, dense image captioning model per-frame, and then linking objects together is poorly suited to this scenario: the captions at each frame are likely to be different due to subtle appearance changes across frames. This motivates our end-to-end trained model, which includes a novel end-to-end tracking algorithm that aggregates features of the same object across time, enabling the subsequent captioner to leverage global features to produce coherent captions.

Although we are the first to our knowledge to study Dense VOC, we can still repurpose existing video grounding datasets for evaluation and domain-specific finetuning. We use VidSTG (Zhang et al., 2020) and VLN (Voigtlaender et al., 2023), originally designed for spatiotemporal sentence grounding: Instead of finding an object tube given a sentence query (grounding), we predict object trajectories directly and use the sentence queries as the ground truth captions. In addition, we show that our generative model trained for Dense VOC can perform grounding by simply

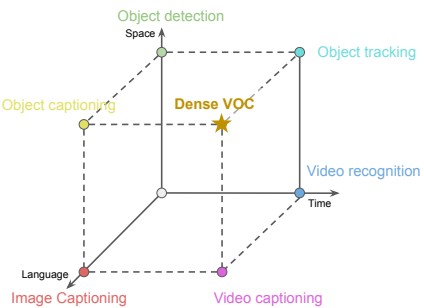

Figure 2: **Overview of Dense VOC**. Our problem involves understanding across space, time, and language, and thus encompasses other vision tasks, which typically consider one or two of these axes. We show these subtasks are complementary, and pretraining on them enables zero-shot generalization to Dense VOC.

selecting the bounding boxes with the maximum likelihood of producing the query sentence. We also develop a new metric that jointly measures captioning, detection and tracking accuracy by extending HOTA (Luiten et al., 2021), the most popular metric for multi-object tracking.

Experiments show that our end-to-end trained Dense VOC model outperforms baselines consisting of strong, per-task models by a substantial margin, producing more accurate and inherently temporally consistent captions. Moreover, we achieve significant improvements from our disjoint, multi-dataset training. We additionally show how we can readily apply our model to related domain-specific datasets: by finetuning our model on a recent person tracking and captioning dataset, BenSMOT (Li et al., 2024b), we outperform prior work by 18.2 points. Furthermore, by applying our generative captioning model to the discriminative grounding task, we are able to outperform dedicated spatial grounding models on both VidSTG and VLN. In summary, we propose the following contributions:

1. We propose the new task of Dense Video Object Captioning. We propose novel evaluation metrics, and repurpose existing grounding datasets for evaluation.

2. We design an end-to-end architecture for our task, with a novel tracking algorithm and feature aggregator that ensures temporally consistent captions. Unlike conventional offline trackers, our tracker is trained end-to-end with the model and produces long-term trajectory features for subsequent captioning.

3. We show our model can be trained without full annotations for the task, with a mixture of disjoint datasets which supervise different parts of our model.

4. We further show how our models generalize to downstream video grounding tasks, achieving state-of-the-art results on two datasets, without explicitly being trained for grounding.

5. Moreover, we significantly improves the state-of-the-art on the BenSMOT dataset Li et al. (2024b) for Semantic Multi-Object Tracking.

## 2  RELATED WORK

**Image captioning** (Chen et al., 2015; Anderson et al., 2018; Xu et al., 2015; Rennie et al., 2017) describes the content of an image with language. State-of-the-art methods map the input image to output text by using multi-modal models (Jiang et al., 2020; Desai & Johnson, 2021; Li et al., 2020; Zhang et al., 2021a; Li et al., 2023; Yu et al., 2022) pretrained on large datasets (Sharma et al., 2018; Radford et al., 2021). For example, GIT (Wang et al., 2022) simple forwards vision tokens from a ViT encoder (Dosovitskiy et al., 2021) to an auto-regressive language decoder (Vaswani et al., 2017; Devlin et al., 2019). Similar ideas apply to **video captioning** (Xu et al., 2016; Zhou et al., 2018; Monfort et al., 2021), by concatenating (Wang et al., 2022) or pooling (Yan et al., 2022) features from each frame, before feeding them to an auto-regressive text decoder. Our work builds on existing captioning architectures (Wang et al., 2022), and extends them to trajectory captioning using our end-to-end model and weak supervision (Monfort et al., 2021; Krishna et al., 2017b; Lin et al., 2014).

**Dense object captioning** in contrast, detects objects in an image and describes them with text (Johnson et al., 2016; Li et al., 2019; Shao et al., 2022; Wu et al., 2022a). It was popularized by the Visual Genome (Krishna et al., 2017b) dataset, which contains full annotations for the task. Early work, DenseCap (Johnson et al., 2016) used a one-stage detector (Redmon et al., 2016) followed by an LSTM text decoder (Hochreiter & Schmidhuber, 1997) on dense feature maps. Most recently, GRiT (Wu et al., 2022a) built upon the state-of-the-art image captioning architecture of GIT (Wang et al., 2022), and generated object captions, also with a transformer decoder (Vaswani et al., 2017), from RoI-pooled (He et al., 2017) image features. Our model advances architectures like GRiT to videos and incorporates end-to-end tracking. We also note that **dense video captioning** in the literature refers to the task of localizing and captioning multiple events *temporally* in videos (Krishna et al., 2017a; Zhou et al., 2018; Wang et al., 2021a; Yang et al., 2023a). Our task, in contrast, involves tracking and captioning objects in a video, and therefore requires *spatial* localization, which is why we name our task "dense video object captioning".

**Multi-object tracking** detects objects and track them with a consistent identity label. The predominant approach is tracking-after-detection (Bewley et al., 2016; Zhang et al., 2021b; Du et al., 2021), *i.e.* first running detectors on each frame and then using a separate tracker to link them. While this works well for existing benchmarks with only a few classes (Dendorfer et al., 2021; Geiger et al., 2012; Yang et al., 2019), it is more challenging in our case: we need tracks *before* captioning to have a single, consistent textual output for the whole trajectory. Thus, our work follows **end-to-end multi-object tracking** (Cheng et al., 2022; Li et al., 2022a; Wang et al., 2021c; Zhou et al., 2022b). We adopt a global tracker GTR (Zhou et al., 2022b), which casts tracking as pairwise association among all objects within a video. Whilst GTR applies a sliding-window-based identity association algorithm during inference as a post-processing step, we design an efficient algorithm to perform this process end-to-end. This is necessary for our task, since our trajectory features are used by a subsequent captioning module which is trained jointly. We are not aware of prior work which efficiently assigns object identities and corresponding features to tracks, and trains end-to-end through this process. Finally, note that **video object tracking and segmentation** (Yang et al., 2021; 2023b; Yang & Yang, 2022; Cheng & Schwing, 2022; Cheng et al., 2024) focuses on following only a *single* object which is given in the first frame (Perazzi et al., 2016; Xu et al., 2018). This is therefore a different setting from our task of detecting, tracking and captioning multiple objects.

**Video object grounding** (Zhang et al., 2020; Voigtlaender et al., 2023) finds a spatio-temporal tube given a video and query sentence as inputs. Existing, discriminative methods (Zhang et al., 2020; Yang et al., 2022; Jin et al., 2022; Su et al., 2021) co-embed visual and text inputs, and use the sentence feature to find the corresponding object. In contrast, we use our generative language model for this task by selecting the object with the highest likelihood of producing the query. To our knowledge, we are the first work to explore the alternate paradigm of generative models for this task. Finally, we note that these tasks are also related to video-referring segmentation (Bellver et al., 2020; Wu et al., 2022b; Yu et al., 2016) which grounds textual queries to segmentation masks. Segmentation, however, is not the focus of our work.

Concurrent to our work, BeyondMOT (Li et al., 2024b) proposes an video object tracking and captioning benchmark and model. We highlight two differences: 1. Li et al. (2024b) uses a frame-by-frame tracker similar to our baselines (Tab. 2), and we propose a novel end-to-end tracker. 2. Our work aims to track and caption **all objects** in the video, while Li et al. (2024b) handles **only**

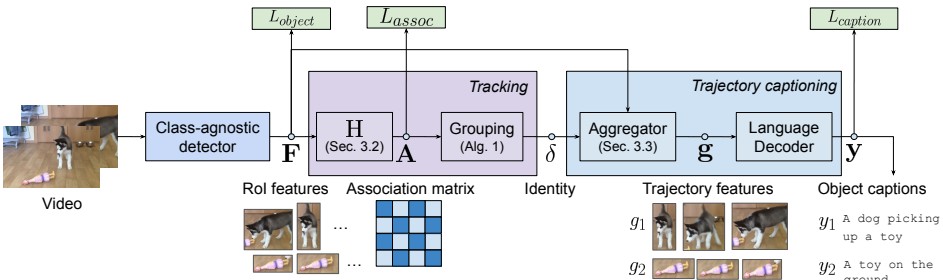

Figure 3: **Overview of our model.** Our end-to-end model has three modules: First it produces object proposals per-frame using a class-agnostic detector (left, trained with detection loss, $L_{object}$). These object proposals are then passed to an end-to-end tracking module that groups objects into trajectories (middle, trained with association loss, $L_{assoc}$). The identities produced by the tracking module are used to aggregate features which are then fed to a language decoder to produce the final caption (right, trained with caption loss $L_{caption}$). Our model can be trained end-to-end with partial supervision on different and disjoint datasets to provide zero-shot Dense VOC capabilities.

**persons**. As a result, our task is much more challenging, and we show our model yields superior performance on their benchmark. OW-VISCap (Choudhuri et al., 2024) on the other hand augments a video segmentation model, Cheng et al. (2022), with a language model (OPT with 2.7 billion parameters (Zhang et al., 2022a)) head for video segmentation and captioning. In contrast, our model is trained flexibly using our disjoint pretraining, which enables us to achieve better detection and tracking performance whilst still using a substantially smaller model.

## 3 METHOD

As shown in Fig. 3, our end-to-end model consists of interlinked heads for object proposal, tracking and captioning the resulting trajectories. Before introducing our novel components, we review prior techniques for captioning and dense object captioning in images (Wu et al., 2022a; Wang et al., 2022).

### 3.1 BACKGROUND

Image captioning maps an input image, $\mathbf{I} \in \mathbb{R}^{H \times W \times 3}$, to a caption $c = (y_1, y_2, \ldots, y_{n_t})$ which is a sequence of up to $n_t$ text tokens from a given vocabulary. The minimal set of components is an image encoder, followed by a text decoder (Vaswani et al., 2017). The encoder maps the input image $\mathbf{I}$, to a feature representation $\mathbf{f} \in \mathbb{R}^{n_v \times d}$ consisting of $n_v$ tokens with dimensionality $d$. The subsequent text decoder is auto-regressive (Graves, 2013) – it predicts the next text token, $y_i$, as a function of both the image features, $\mathbf{f}$, and previously generated text tokens, $\mathbf{y}_{0:i-1}$, denoted by $y_i = \text{Decode}(\mathbf{f}, \mathbf{y}_{0:i-1})$. Note that the first step of decoding begins with $y_0 = \text{BOS}$, a special beginning-of-sentence token, and the caption ends when the end-of-sentence token, $\text{EOS}$, is output by the model. This simple image captioning model has been demonstrated to be effective and scalable by GIT (Wang et al., 2022), achieving state-of-the-art results across a number of captioning datasets.

GRiT (Wu et al., 2022a) extends the approach further to dense object captioning of images: Here, the authors use an object proposal network (Zhou et al., 2019) to produce a set of $K$ class-agnostic bounding boxes, $b_1, b_2, \ldots, b_K$. Features corresponding to each of these objects are obtained using RoIAlign (He et al., 2017), resulting in a localized feature, $f_k \in \mathbb{R}^{r \times r \times d}$ where $r = 7$ is the output resolution of RoIAlign. Each of these grid features is flattened into $f_k \in \mathbb{R}^{r^2 \times d}$ and decoded independently by the text decoder, as done in GIT. Therefore, the loss used to train a GRiT model consists of $L = L_{object} + L_{caption}$ where $L_{caption}$ is a cross-entropy loss over all text tokens in the vocabulary, and $L_{object}$ consists of bounding box regression and objectness terms, as standard in object detection literature (Zhou et al., 2019; Ren et al., 2015; Lin et al., 2017).

We now describe how we extend object captioning to videos by tracking object proposals over time (Sec. 3.2) and aggregating trajectory features and captioning them (Sec. 3.3) in an end-to-end fashion. Section 3.4 explains how we train our model, whilst Sec. 3.5 describes how we apply our model directly to video object grounding tasks.

### 3.2 END-TO-END TRACKING

As shown in Fig. 3 (left), we first produce object proposals separately for each frame. Tracking then aims to assign each object in each frame a unique trajectory identity $\delta \in \mathbb{N}$. We define $\mathbf{f}_k^t \in \mathbb{R}^{r^2 \times d}$ as the ROI feature of object proposal $k$ in frame $t$, $\mathbf{F} = [\mathbf{f}_k^t]_{t=1,k=1}^{T,K_t}$ as the concatenation of all object

---

**Algorithm 1:** Identity assignment from association matrix. This greedy algorithm can be implemented efficiently on accelerators, enabling end-to-end training.

---

**Input** : Association Matrix $\mathbf{A} \in \mathbb{R}^{\mathbf{TK} \times \mathbf{TK}}$ // $T$: num. frames. $K$: num. objects per frame.
**Hyperparameters** : Association score threshold $\theta$
**Output** : Identities for each object $\delta \in \mathbb{N}^{TK}$
$M \leftarrow T \times K$           // Number of total objects.
$A \leftarrow preprocess(A)$       // Preprocess $A$ to ensure object pairs in the same frame have a score of 0.
$\hat{A} \leftarrow (A \geq \theta).astype(bool)$       // Binary matrix for possible merges.
$\delta \leftarrow zeros(M)$       // Initialize output identities, shape $(M, )$
$id\_count \leftarrow 0$       // Initialize ID count.
**while** $\hat{A}.any() > 0$ **do**
     $track\_len \leftarrow \hat{A}.sum(axis=1)$       // Number of objects in each merge.
     $i \leftarrow track\_len.argmax()$       // Find the longest track to merge.
     $id\_count \leftarrow id\_count + 1$       // Create a new identity.
     $\delta \leftarrow \delta + id\_count * \hat{A}_i$       // Assign the current track a new ID using $\hat{A}_i$.
     $\hat{A} \leftarrow \hat{A} - \hat{A}_{i\cdot} | \hat{A}_{\cdot i}$       // Remove merged indices. "|" is logical or.
**end**
**return** $\delta$

---

features in the video. Let $M = |\mathbf{F}| = \sum_{t=1}^{T} K_t$ as the total number of objects in all frames, where $K_t$ is the number of object proposals at the $t^{\text{th}}$ frame. Thus, we have $\mathbf{F} \in \mathbb{R}^{M \times r^2 \times d}$.

From these object features, $\mathbf{F}$, we predict a global association matrix, $\mathbf{A} \in \mathbb{R}^{M \times M}$, where $\mathbf{A}_{ij} = 1$ if the objects denoted by the $i^{th}$ row and $j^{th}$ column are from the same trajectory (Fig. 3 middle). Otherwise, $\mathbf{A}_{ij} = 0$ means that they are from different trajectories, or one of them is the background.

We use a transformer module, H, with two self-attention layers, similar to Zhou et al. (2022b), to predict the association matrix $\mathbf{A} = \sigma(\text{H}(\mathbf{F}))$, where $\sigma$ is the sigmoid activation. Given the object trajectory annotations, we construct the ground truth association matrix $\bar{\mathbf{A}}$ for $\mathbf{A}$, where $\bar{\mathbf{A}}_{ij} = 1$ if and only if row $i$ and column $j$ of $\mathbf{A}$ are matched to the same ground truth trajectory using an Intersection over Union (IoU) criteria of 0.5. The training loss $L_{assoc}$ for this module is then a binary cross entropy between $\mathbf{A}$ and $\bar{\mathbf{A}}$, $L_{assoc} = \frac{1}{M} \sum_{ij} \text{BCE}(A_{ij}, \bar{A}_{ij})$.

After constructing our association matrix, $\mathbf{A}$, we need to aggregate object-level features according to identities $\delta = [\delta_k^t]_{t=1, k=1}^{T, K_t}$, to generate trajectory-level captions for the next captioning stage. Here, $\delta_k^t$ denotes the identity of the $k$-th object proposal in the $t$-th frame. We design a greedy grouping algorithm (Alg. 1) operating on $\mathbf{A}$ to obtain $\delta$. Concretely, we greedily extract the longest trajectory from untracked objects, until there are no possible associations left (indicated by the association score being above a threshold $\theta$). This guarantees each trajectory has at most one object in each frame. This algorithm can be implemented efficiently on accelerators, allowing us to backpropagate through it.

As aforementioned, prior trackers (Zhang et al., 2021b; Zhou et al., 2020; 2022a) do not explicitly perform identity assignment within the model, but rather as a post-processing step since tracking is the final output for such methods. Our work efficiently assigns object identities to tracks in an end-to-end trainable network, which enables us to perform joint trajectory-level captioning training as described next.

## 3.3 TRAJECTORY CAPTIONING

Our end-to-end tracking module produces object features, $\mathbf{f}_k$ (we omit the frame index $t$ below for clearer notation), paired with their identities, $\delta_k$, which denote their correspondence over time. We now describe two methods for aggregating features along this trajectory in order to caption it.

**Soft aggregation.** A straightforward way to leverage object features over time is to compute a weighted sum to combine them into a single, global trajectory feature. We observe that the association matrix, $\mathbf{A}$ (Sec. 3.2), already serves as a summation weight. Specifically, we set $\mathbf{G} = \frac{\mathbf{A}}{||\mathbf{A}||} \cdot \mathbf{F}$, where $\cdot$ denotes matrix multiplication, and $|| \cdot ||$ normalizes $\mathbf{A}$ by rows. Each row of $\mathbf{G}$, $\mathbf{g}_k \in \mathbb{R}^{r^2 \times d}$, therfore denotes an aggregated feature over its trajectory for object $k$.

**Hard aggregation.** An alternative to weighted temporal averaging is to concatenate and construct new trajectory features. Let $\mathbf{f}_\tau = \{\mathbf{f}_{k'}\}_{\delta_{k'} = \tau}$ be the set of all object features with identity $\tau$. We note $\mathbf{f}_\tau$ can be as long as the entire video, and thus it may be expensive to directly use $\mathbf{f}_\tau$. Therefore, we uniformly

| Dataset | Annotation type | Train set size ($10^3$) | $L_{object}$ | $L_{assoc}$ | $L_{caption}$ |
|---|---|---|---|---|---|
| COCO (Lin et al., 2014) | Image detection | 118 | ✓ | | |
| VG (Krishna et al., 2017b) | Image object captioning | 70 | ✓ | | ✓ |
| SMiT (Monfort et al., 2021) | Video captioning | 480 | | | ✓ |
| Aug-COCO (Lin et al., 2014) | Video object tracking | 118 | ✓ | ✓ | |

Table 1: **Datasets for pretraining.** We supervise different losses based on available annotations.

sample a subset of object features from the trajectory, denoted as $\mathbf{g}_\tau = \text{UniformSample}(\mathbf{f}_\delta, m)$, where $\mathbf{g}_\tau \in \mathbb{R}^{mr^2 \times d}$, inspired by Wang et al. (2022). $m$ is the number of sampled frames, and we set $m = 6$ following ablations in Appendix C.2.

The trajectory-aggregated features for each object, $\mathbf{g}_k$, are then autoregressively decoded into output captions for each object, $\mathbf{y}_k$. This follows Sec. 3.1, where $y_{k,i} = \text{Decode}(\mathbf{g}_k, \mathbf{y}_{k,0:i-1})$. Note that the language decoder has the same parameters as in single-frame object captioning, but processes more input tokens. Therefore, we train it in the same manner with a softmax cross-entropy loss over the vocabulary of text tokens, denoted by $L_{caption}$.

### 3.4 Pretraining with disjoint subtasks

As shown in Fig. 3, our model is trained with the loss function, $L = L_{object} + L_{assoc} + L_{caption}$. When we have full Dense VOC annotations, which supervise each component of our model we can train our entire model end-to-end. However, to leverage more weakly-labeled data, we can also decompose Dense VOC into subtasks, and use each subtask to supervise the relevant part of our model using the available annotations as shown in Tab. 1. This approach also enables us to perform our final task in a zero-shot manner (*i.e.* without training on any full Dense VOC annotations).

**Object detection.** Using detection datasets for images, we can train the object proposal generator with $L_{object}$. We use COCO (Lin et al., 2014) as it is the most popular dataset for this task.

**Dense captioning in images.** Dense object captioning datasets of images allow us to train both the object proposal generator and the text decoder, by supervising $L_{object}$ and $L_{caption}$. Here, we use Visual Genome (Krishna et al., 2017b), the largest dataset for this task.

**Global video captioning.** Video captioning datasets help us to reduce the domain gap to our final task by also training on video. In particular, we use Spoken Moments in Time (SMiT) (Monfort et al., 2021) which is the largest dataset for this task and contains narrations for short clips (roughly 3 seconds). As there are no object annotations, but only video-level captions, we construct an object proposal from the entire frame and caption that with our text decoder, applying $L_{caption}$. This approach is inspired by prior work on weakly-supervised object detection (Zhou et al., 2022a; Bilen & Vedaldi, 2016; Arnab et al., 2020).

**Tracking.** Training the tracking module of our network (Sec. 3.2) requires annotations that associate detections of an object identity throughout the video. We found that existing tracking datasets either have too limited vocabularies for general objects (MOT (Dendorfer et al., 2021), KITTI (Geiger et al., 2012), YouTube VIS (Yang et al., 2019)), or are too small (TAO (Dave et al., 2020) and UVO (Wang et al., 2021b) label 600 and 5 000 videos respectively), and thus giving unsatisfactory results in our setting (Appendix C.3). As a result, following existing work (Zhang et al., 2021b; Zhou et al., 2020), we instead augment image datasets into tracking ones by applying two different data augmentations to the same image, and then linearly interpolating the frames in between to form a pseudo-video. In particular, we augment COCO (referred to as Aug-COCO (Zhou et al., 2020)). This enables us to apply $L_{assoc}$ and $L_{object}$ when training our model.

### 3.5 Application to video object grounding

The task of video object grounding (Zhang et al., 2020; Voigtlaender et al., 2023) consists of two inputs: a video, $\mathbf{V}$, and a sentence query, $\bar{c}$. The output is a sequence of bounding boxes, $[b^s, b^{s+1}, \ldots, b^e]$, corresponding to the sentence query, where $s$ and $e$ are the indices of the start and end frames respectively.

Our model, however, generates captions, $c$, at the output, rather than requiring it as an input. To apply our model to grounding, we follow an analogous approach to prior works that performed closed-set image classification with captioning models (Alayrac et al., 2022; Chen et al., 2023b): we evaluate the likelihood (i.e., exponential negative cross-entropy loss) of the sentence query, $\bar{c}$, for each of the object trajectories produced by our model. In practice, we find that instead of just taking the

object trajectory with the highest sentence-likelihood, we achieve higher accuracy by weighting the likelihood by the detection score, $s_k$, from our object proposal module. Thus, given bounding boxes, trajectory features and detection scores, $\{(b_k^t, s_k^t, \mathbf{g}_k)\}_{t=1,k=1}^{T,K_t}$, we choose the bounding boxes with the highest weighted likelihood:

$$k^* = \arg\max_k \left( s_k^t \cdot \exp(-L_{caption}(\text{Decode}(\mathbf{g}_k), \bar{c})) \right), \qquad b^t = b_{k^*}^t. \tag{1}$$

## 4 EXPERIMENTAL EVALUATION

As we are proposing a new task, there is no dedicated dataset or evaluation metric for dense video object captioning for all objects. Fortunately, existing video grounding datasets (Zhang et al., 2020; Voigtlaender et al., 2023) have annotations for object trajectories and their captions, allowing us to repurpose them for Dense VOC, as defined next. We also report results on concurrent person-focused video object tracking and captioning benchmark, BenSMOT(Li et al., 2024b).

### 4.1 DATASETS

**VidSTG (Zhang et al., 2020)** was originally created for spatio-temporal sentence grounding, but can be used for Dense VOC: Each video annotates multiple textual queries and their corresponding spatio-temporal tubes. By aggregating these across all videos, we obtain the paired trajectory-caption annotations that we need for training and evaluating our model.

VidSTG has exhaustive trajectory (*i.e.* bounding box and tracking) annotations for all objects (Shang et al., 2019), but not all objects are used in grounding, and thus not all objects have captions. We account for this fact in both training and testing. Specifically, we do not compute $L_{caption}$ on objects without caption annotations, and also exclude them during evaluation (see Sec. 4.2). In particular, when a prediction is matched to a ground truth without caption annotations, we do not evaluate its captioning metrics, but still evaluate detection metrics. The dataset contains 5,436 training videos and 602 validation videos, with each video being at most 200 frames long. We use the declarative annotations from the dataset containing 19,000 captioned trajectories for training, and report results on the declarative validation set.

**Video Localized Narratives (VLN) (Voigtlaender et al., 2023)** augments existing datasets by narrating the "actors" in a video. We therefore use these narrations as our target captions. We use the subset from the UVO dataset (Wang et al., 2021b) as UVO has exhaustive detection and tracking annotations for all objects. Like VidSTG, the captions are not exhaustive for all objects, so we exclude objects without captions in both training and evaluating the captioning module. Each video has bounding box annotations for 3 sparsely sampled frames, and thus we train and evaluate on these frames. The dataset contains a total of 5,136 training and 2,451 validation videos.

**BenSMOT (Li et al., 2024b)** contains person bounding boxes trajectories and their manually-annotated captions for 3292 YouTube videos. The dataset has in average 2.2 trajectories per video.

### 4.2 EVALUATION METRICS

**Captioned-HOTA (CHOTA).** Our primary metric, CHOTA, builds on Higher Order Tracking Accuracy (HOTA) (Luiten et al., 2021) – which is now the most popular metric in multi-object tracking – by adding a captioning term. HOTA decomposes tracking into two subproblems: detection and association, with the final score being the geometric mean of detection accuracy (DetA) and Association Accuracy (AssA): $\text{HOTA} = \sqrt{\text{DetA} \cdot \text{AssA}}$. Here, $\text{DetA} = \frac{|TP|}{|TP|+|FP|+|FN|}$, and AssA averages the "Association IoU" over true-positives, as $\text{AssA} = \frac{1}{|TP|}(\sum_{(x,y) \in TP} \text{Ass-IoU(x, y)})$, where $(x, y)$ are the matched prediction-ground truth box pairs in each frame. Note that HOTA computes the DetA and AssA for each detection in each frame, rather than for each trajectory, as the overall trajectory performance is implicitly measured by the association of detections over time. Moreover, it considers all possible trajectory matches that can be made simultaneously (Sec. 7 of Luiten et al. (2021)).

Our task consists of captioning, detection and association. Therefore, we also define an additional "Captioning Accuracy" (CapA) term as:

$$\text{CapA} = \frac{1}{3|TP'|} \sum_{(x,y) \in TP'} (\text{METEOR}(x, y) + \text{CIDEr}(x, y) + \text{SPICE}(x, y)), \tag{2}$$

| # | CHOTA | DetA | AssA | CapA | Consistent captions |
|---|---|---|---|---|---|
| 1 Per-frame cap. w. IOU tracker | 49.9 | **64.4** | 52.2 | 37.1 | ✗ |
| 2 Per-frame cap. w. FairMOT (Zhang et al., 2021b) | 51.2 | 63.4 | 57.2 | 37.0 | ✗ |
| 3 Per-frame cap. w. ByteTrack (Zhang et al., 2022b) | 52.3 | 64.2 | 60.2 | 37.1 | ✗ |
| 4 Middle-frame cap. w. ByteTrack (Zhang et al., 2022b) | 50.7 | 64.2 | 60.2 | 33.8 | ✓ |
| 5 Ours, soft aggregation | 54.6 | **64.4** | **65.9** | 38.4 | ✓ |
| 6 Ours, hard aggregation | **54.9** | 64.2 | **65.9** | **39.1** | ✓ |

Table 2: **Comparison of our end-to-end model to per-task baselines on VidSTG validation.** Our models are based on #2 of Tab. 3 right. The image dense captioning models used in the baselines (rows #1-#4) are trained on the same datasets, and run off-the-shelf trackers as post-processing. Our end-to-end approach improves across all metrics, and produces temporally consistent captions.

which uses three popular image-captioning metrics (Chen et al., 2015), and $TP'$ are the true-positive detection pairs that have caption annotations (as discussed in Sec. 4.1). Note that for compatibility with HOTA, we follow DetA and AssA and compute CapA separately per-object on each frame. The final metric is then CHOTA $= \sqrt[3]{\text{DetA} \cdot \text{AssA} \cdot \text{CapA}}$, effectively adding a captioning term to the HOTA metric. We include further details and code in Appendix B, along with results using the image dense object captioning metrics, mAP-METEOR (Appendix C.1).

## 4.3 IMPLEMENTATION DETAILS

Our implementation is based on the public release of GRiT (Wu et al., 2022a). GRiT uses a ViTDet-Base (Li et al., 2022b) backbone (initialized with CLIP (Radford et al., 2021)), a CenterNet (Zhou et al., 2019) object proposal network and RoI Head, and a randomly-initialized text decoder.

We first train our model for general Dense VOC on large-scale disjoint datasets (Sec. 3.4). During disjoint pretraining, we sample batches from different datasets with an even ratio, (1: 1: 1: 1), thus avoiding additional hyperparameters. For video datasets, we sample 8 frames for a video and use a local batch size of 1. For image datasets, we use a local batch size of 8. We train our model on 32 GPUs, which means we have an effective batch size of 256 images or 32 videos.

We evaluate the models on the two fully-annotated datasets (Sec. 4.1) in zero-shot and full-finetuning setups. For VidSTG, we sample 16 frames during training, and then run on all 200 frames during testing. For VLN, we use all 3 annotated frames in both training and evaluation. In both cases, we use an input size of $384 \times 384$. During inference, we threshold the outputs of our object proposal module with a score of $0.5$, and only track the remaining objects. We include exhaustive implementation details and hyperparameters in Appendix B.2 with the full code. We report runtime in Appendix B.3.

## 4.4 ANALYSIS OF END-TO-END TRACKING

We first study the benefits of our end-to-end model in Tab. 2. We do this by comparing to multiple, strong baseline models running in sequence. Concretely, we use the state-of-the-art image-dense object captioning model Wu et al. (2022a) followed by tracking as a post-processing step. We use trackers ranging from a simple IoU-based tracker Wu et al. (2019) to more recent, sophisticated methods like FairMOT (Zhang et al., 2021b), and ByteTrack (Zhang et al., 2022b).

As the baseline predicts captions independently on each frame, the caption is not consistent over the entire trajectory. Therefore, we consider an additional baseline where we only use the caption from the middle frame of the trajectory. Finally, note that as our baseline captioner is pretrained on Visual Genome, and then finetuned on individual frames of VidSTG, it has been trained on identical data to our model, allowing us to make fair comparisons.

As shown in Tab. 2, per-frame captioners followed by offline trackers produce temporally inconsistent captions (#1-#3). Naively selecting the caption from the middle frame as the trajectory-level caption produces temporally consistent captions, but comes at the cost of captioning accuracy, as a single frame may not be representative of the entire event (#4). Both variants of our model (#5 and #6) improve tracking quality substantially, as shown by their large improvement on AssA, demonstrating the benefits of end-to-end training and incorporating temporal information. Our model improves on CapA too, showing that improved object trajectories provide better features for subsequent captioning. Finally, we note that the quality of the initial detections at each frame, measured by DetA, does not really change between the baselines and our method. This does, however, show that training our model jointly with multiple loss functions does not compromise performance on individual tasks.

Overall, our end-to-end model (#6) improves the CHOTA by 2.6 points over the best baseline (#3). As hard aggregation performs slightly better, we use it in our following experiments.

| # | COCO | VG | SMiT | Aug COCO | VidSTG (zero-shot) | | | | VLN (zero-shot) | | | | VidSTG (finetuned) | | | | VLN (finetuned) | | | |
|---|------|----|------|----------|-------|------|------|------|-------|------|------|------|-------|------|------|------|-------|------|------|------|
| | | | | | CHOTA | DetA | AssA | CapA | CHOTA | DetA | AssA | CapA | CHOTA | DetA | AssA | CapA | CHOTA | DetA | AssA | CapA |
| 0 | | | | | - | - | - | - | - | - | - | - | 47.8 | 54.6 | 57.8 | 34.5 | 29.7 | 35.3 | 85.4 | 8.7 |
| 1 | ✓ | | | | - | 48.9 | - | - | - | 27.8 | - | - | 52.3 | 64.9 | 63.0 | 34.9 | 31.8 | 43.9 | 88.7 | 8.2 |
| 2 | | ✓ | | | - | 17.8 | - | 7.8 | - | 12.1 | - | 7.4 | 54.9 | 64.2 | 65.9 | 39.1 | 40.6 | **45.1** | 88.4 | 16.7 |
| 3 | | | ✓ | | - | - | - | - | - | - | - | - | 45.4 | 51.9 | 56.9 | 31.6 | 37.4 | 41.2 | 87.7 | 14.5 |
| 4 | | ✓ | ✓ | | - | 19.1 | - | 8.5 | - | 14.3 | - | 8.5 | 55.2 | 64.0 | 67.1 | 39.2 | 41.0 | 44.2 | 88.4 | **17.8** |
| 5 | ✓ | ✓ | | | - | 49.9 | - | 8.1 | - | 28.0 | - | 7.8 | 55.6 | 65.7 | 68.9 | 38.4 | 40.9 | 44.1 | 88.8 | 17.4 |
| 6 | ✓ | | ✓ | | - | 50.4 | - | 4.9 | - | 28.7 | - | 7.5 | 54.4 | 64.9 | 63.9 | 38.8 | 35.6 | 43.7 | 88.5 | 11.6 |
| 7 | ✓ | ✓ | ✓ | | - | 51.3 | - | 9.1 | - | **29.9** | - | 9.0 | 56.5 | **65.8** | 68.2 | **40.1** | 41.1 | 44.2 | 88.9 | 17.7 |
| 8 | ✓ | ✓ | ✓ | ✓ | 31.1 | 51.4 | 59.6 | 9.8 | 29.2 | 29.1 | 88.0 | 9.7 | 56.9 | 65.8 | 70.4 | 39.7 | 41.3 | 44.3 | 89.5 | 17.7 |

Table 3: **Zero-shot (left) and finetuning (right) evaluation of our disjoint trained models with varying datasets.** We show results on VidSTG (Zhang et al., 2020) and VLN (Voigtlaender et al., 2023). Each row is a model pretrained on the specified datasets for zero-shot evaluation and then finetuned on the downstream datasets. #0 is finetuned from a CLIP checkpoint. For models without tracking supervision (#1–7), we cannot report their zero-shot association accuracy (AssA). Our full model (#8) gains full Dense VOC ability from disjoint training, and shows good performance on all metrics with or without finetuning, on both datasets. Detailed captioning metrics are in Appendix C.4.

## 4.5 ANALYSIS OF DISJOINT TRAINING

**Zero-shot evaluation.** We first pretrain on multiple disjoint datasets (Sec. 3.4), and evaluate zero-shot on our target datasets, VidSTG and VLN, without training on them in Tab. 3 left. Zero-shot evaluation is simple to perform for captioning models compared to classification, thanks to their open vocabulary.

As mentioned in Tab. 1 and Fig. 3, each dataset supervises different parts of our model. For example, a model that is only trained on COCO (#1 in Tab. 3), is only trained with $L_{object}$, meaning that it only produces object proposals which we can evaluate with the Detection Accuracy component of our CHOTA metric. Visual Genome (VG) can supervise both the object proposal and captioning heads of our model. However, there is a large domain gap between the captions in VG and our target datasets, since the captions in VG are for single images and focus on very different vocabularies. Furthermore, VG tends to annotate bounding boxes around object parts rather than entire objects. Consequently, our zero-shot DetA is low when training only on VG (#2). To mitigate the differences in the type of bounding boxes annotated by VG, we ignore $L_{object}$ on it when using it in conjunction with COCO. Note that we cannot evaluate a model trained only on SMiT, as it does not produce bounding boxes.

We observe in Tab. 3 (left) that the different datasets have complementary properties: Adding COCO improves detection accuracy (#2 to #5, #4 to #7), and adding SMiT improves the captioning accuracy (#2 to #4, #5 to #7) even though SMiT only captions at a video-level. Finally, training with Aug-COCO allows us to also supervise $L_{assoc}$ and thus the tracking module of our model. A model trained on all the datasets (#8) can therefore perform the full Dense VOC task, and shows good performance on all individual metrics compared to models trained on fewer datasets. Notably, we observe our final model with tracking improves captioning ability (CapA) without adding captioning training data. Similar to Tab. 2, the improvements are likely from our ability to leverage temporal information.

**Finetuning evaluation.** We now finetune each of the pretrained models from Tab. 3 left and show results in the right. We also include a baseline (#0) which initializes from only a CLIP-pretrained checkpoint, observing that this model performs poorly. Once again, we observe that different pretraining datasets are complementary, as adding either SMiT or COCO (#2 to #4, #2 to #5, #1 to #6) improves results further. Adding more pretraining datasets improves results further (#7), and we achieve the best results with our model pretrained on all pretraining datasets (#8), which outperforms the best single-dataset pretrained model by 2.0 CHOTA on VidSTG, and 0.7 CHOTA on VLN. The improvement over only a CLIP-pretrained checkpoint is even larger, by 9.1 CHOTA and 11.6 CHOTA on the two respective datasets. Qualitative visualizations are shown in the supplement.

## 4.6 COMPARISON TO CONCURRENT WORKS BENSMOT AND OW-VISCAP.

We compare to the concurrent work Li et al. (2024b), which focuses on person category rather than all classes like our task. We finetune our model on the training set using the same hyper-parameters as our VidSTG experiments. Our full model achieved 90.19 HOTA and 0.254 CIDEr on this benchmark, significantly outperforming Li et al. (2024b). There are two major advantages of our model: our disjoint pretraining, and our use of a larger backbone (ViT-B vs. DLA-34 in BeyondMOT). We further break down the improvements by removing these two components in Tab. 4. The results show: 1. Our pretraining provides consistent gains on Li et al. (2024b) benchmark, improving especially captioning metrics. 2. With a small backbone and no pretraining, our model still outperforms Li et al. (2024b) on tracking and captioning metrics, showing the advantages of our end-to-end architecture.

|  | HOTA | DetA | AssA | CIDEr |
|---|---|---|---|---|
| Li et al. (2024b) | 71.98 | 80.79 | 73.71 | 0.087 |
| Ours | **90.19** | **90.79** | **89.59** | **0.254** |
| - Pretrain | 88.56 | 89.38 | 87.74 | 0.150 |
| - Backbone | 86.55 | 84.33 | 89.19 | 0.129 |

Table 4: **State-of-the-art comparison on BenS-MOT (Li et al., 2024b).** Our model outperforms Li et al. (2024b) on comparable setting (no extra data, small backbone), and our full model improved 18.2 HOTA and 0.167 CIDEr.

|  | CHOTA | DetA | AssA | CapA |
|---|---|---|---|---|
| OW-VisCap | 53.0 | 60.1 | 54.0 | 43.9 |
| Ours | 56.9 | 65.8 | 70.4 | 39.7 |

Table 5: **Compare with concurrent work OW-VisCap (Choudhuri et al., 2024) on VidSTG.** The results are from Choudhuri et al. (2024) Tab. 2 and our Tab. 3 #8 under the same setting. Our model are better at detection and tracking, with lower captioning accuracy due to a smaller langauge head (46M vs. 2.7B params.).

|  | Backbone | Rec. & Prec.$> 0.5$ |
|---|---|---|
| ReferFormer | ResNet50 | 48.3 |
| GRiT | ViT-B | 62.1 |
| Ours | ViT-B | **65.1** |

Table 6: **State-of-the-art comparison of spatial grounding on VLN Location-QA Voigtlaender et al. (2023).** We report the official metric, which evaluates if bounding box recall and precision are both above 0.5. We compare to the ReferFormer baseline Voigtlaender et al. (2023), GRiT Wu et al. (2022a), and our model (#8 of Tab. 3 left).

|  | Finetuned | Zero-shot |
|---|---|---|
| STVGBert | 47.3 | - |
| TubeDETR | 59.0 | - |
| STCAT | 61.7 | - |
| Ours | **61.9** | 54.1 |

Table 7: **State-of-the-art comparison of spatial grounding on the VidSTG** with STVGBert Su et al. (2021), TubeDETR Yang et al. (2022), and STCAT Jin et al. (2022). All models use ground truth temporal localization.

We compare to OW-VISCap which uses a Mask2Former architecture with video object queries. Tab. 5 shows an improved overall performance in CHOTA. Our largest improvement is in Association Accuracy, showing that our end-to-end tracking module (Sec. 3.2) outperforms the Mask2Former counterparts. OW-VisCap gets a higher captioning accuracy as they used a substantially larger, 2.7 billion parameter OPT language decoder (Zhang et al., 2022a), whilst we used a smaller, 46 million parameter language model as in GIT (Wang et al., 2022).

### 4.7 STATE-OF-THE-ART COMPARISON ON VIDEO GROUNDING

As introduced in Sec. 3.5, Dense VOC models can be directly used for sentence grounding, by finding the proposals with the maximum likelihood of generating the query. We evaluate spatial grounding on the VLN Location-QA (Voigtlaender et al., 2023) and VidSTG (Zhang et al., 2020) benchmarks.

**VLN Location-QA** consists of questions starting with "Where is", and requires the model to produce a bounding box at each frame in the video. The task is therefore effectively a sentence grounding problem, and indeed, the ReferFormer (Wu et al., 2022b) baseline used by Voigtlaender et al. (2023) performs sentence grounding after removing "Where is" from the question. We also remove this prefix before grounding following Sec. 3.5 for both our final model, and an additional GRiT baseline.

In this dataset, only one annotated frame (unknown at inference time) is evaluated, and this benchmark therefore effectively does not involve temporal localization. As the annotation of this dataset is based on mouse traces instead of bounding boxes, the evaluation metric considers bounding box coverage (recall) and precision (full details in Voigtlaender et al. (2023)). As shown in Tab. 6, we improve substantially over ReferFormer and our GRiT (Wu et al., 2022a) baseline.

**VidSTG** requires producing a sequence of bounding boxes for a given sentence query. The evaluation metric is the average of the Intersection over Union (IoU) at each frame, between the predicted and ground truth bounding boxes for the target object. We compare to other prior works on this dataset in Tab. 7, assuming that the input video is already trimmed temporally to the objects of interest. Our model achieves the best IoU, outperforming models designed specifically for grounding, thereby showing that our generative framework can be used effectively in the discriminative grounding task. We also evaluate zero-shot without training on VidSTG, and still perform competitively. This emphasizes the efficacy of our disjoint pretraining. We provide more results in Appendix C.5.

## 5 CONCLUSION

We proposed the new task of dense video object captioning. Although this task requires expensive annotations across space, time and language, we show that we can train a model on existing larger-scale datasets for disjoint subtasks. We show our proposed end-to-end architecture is important for producing more accurate and coherent captions.

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

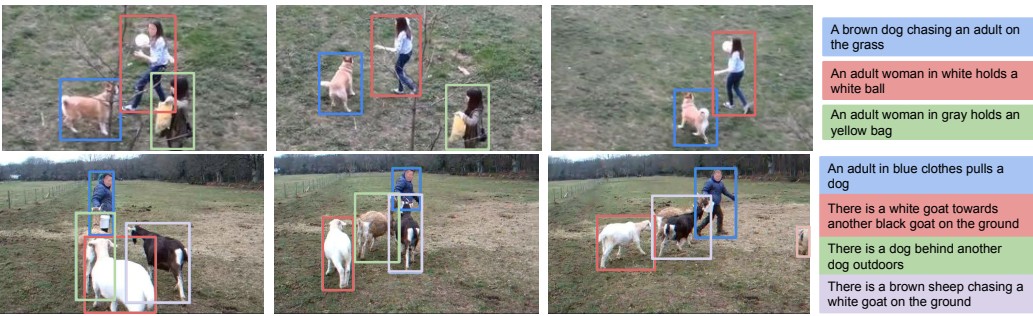

Figure 4: **Qualitative results on VidSTG.** Our model captures motion (1st row) and handles crowded scenes (2nd row). However, it may misrecognize objects (2nd row, "dog" should be "goat") and action boundaries (2nd row, "chasing" before it occurs).

## APPENDICES

We present further qualitative results (App. A), additional experimental details (App. B), additional experimental analysis (App. C), and broader impact and potential negative impact (App. E).

## A  QUALITATIVE RESULTS

We show example qualitative visualizations Fig. 4 and discuss typical failure cases.

## B  ADDITIONAL EXPERIMENTAL AND IMPLEMENTATION DETAILS

### B.1  CODE

Our code is available at https://github.com/google-research/scenic.

Our CHOTA evaluation code is in file "`code/chota.py`". This evaluation code is based on the official HOTA implementation[*]. The original code is under an MIT license.

### B.2  FULL TRAINING DETAILS

As mentioned in Sec. 4.3, our model is based on GRiT Wu et al. (2022a). The original GRiT code[*] is released under an MIT license. Following GRiT, we use a ViTDet-Base Dosovitskiy et al. (2021); Li et al. (2022b) backbone, a CenterNet Zhou et al. (2019) region proposal network and RoI Head, and a randomly-initialized text decoder following that of GIT Wang et al. (2022). The text decoder consists of 6 self-attention layers with casual feature masks Wang et al. (2022). All model architecture parameters follow the defaults from GRiT Wu et al. (2022a).

The original GRiT uses an MAE pretrained checkpoint, while in our case we found a CLIP pretrained checkpoint Radford et al. (2021) performs better on our task. To fit more frames into memory for both training and evaluation, we use a $384 \times 384$ input size instead of the original $1024 \times 1024$. This choice moderately decreases dense image captioning performance on Visual Genome (from 17.3 $\text{AP}_M$ to 15.7 $\text{AP}_M$).

During disjoint multi-dataset pretraining, we sample batches from different datasets in an even ratio $(1 : 1 : 1 : 1)$. For image datasets, a batch is composed of different images; for video datasets, we put the time dimension in batches and always guarantee images in the same mini-batch are from the same video. We use a local batch size of either 1 video (consisting of 8 sampled frames), or 8 images. As we use 32 GPUs, this means that our global batch size is either 32 videos or 256 images. We use the AdamW optimizer with a learning rate of $2 \times 10^{-4}$, weight decay of $0.05$, and a layerwise learning rate decay of $0.7$ Li et al. (2022b); Wu et al. (2022a). We train for $22.5 \times 10^3$ iterations per dataset, decreasing the learning rate by a factor of 10 after $90\%$ and $97.5\%$ of the training schedule Wu et al.

---

[*]https://github.com/JonathonLuiten/TrackEval
[*]https://github.com/JialianW/GRiT

(2022a). For pretraining on all the 4 datasets in Sec. 3.4, this corresponds to a total of $90 \times 10^3$ iterations, which took approximately 20 hours on 32, 16GB V100 GPUs.

For VidSTG Zhang et al. (2020) finetuning, we sample 16 frames in training, and run on all 200 frames in testing. For VLN Voigtlaender et al. (2023) finetuning, we use the 3 annotated frames in both training and evaluation. For finetuning experiments on both datasets, we use a video batch size 16 and train for $11.25 \times 10^3$ iterations, with a learning rate of $10^{-5}$, weight decay of 0.05, and layerwise-learning decay of 0.7 Li et al. (2022b). The finetuning took approximately 6 hours on 16, 16GB GPUs for VidSTG, and about 2 hours on 16, 16GB GPUs for VLN. Inference on VidSTG requires 32GB of GPU memory to fit 200 frames.

**Training losses.** Training our model involves a detection loss $L_{object}$, a tracking loss $L_{assoc}$, and a captioning loss $L_{caption}$, that is

$$L = L_{object} + L_{assoc} + L_{caption}. \tag{3}$$

For completeness, we detail these three terms next:

The detection loss Zhou et al. (2019) involves a center heatmap loss, a bounding box regression loss, and a classification and bounding box refinement loss in the RoI head:

$$L_{object} = L_{heatmap} + L_{reg} + L_{roi\text{-}cls} + L_{roi\text{-}reg}. \tag{4}$$

The heatmap loss is defined on the predicted heatmap $Y \in \mathbb{R}^{H \times W}$ and the ground truth heatmap $\bar{Y} \in \mathbb{R}^{H \times W}$:

$$L_{heatmap}(Y, \bar{Y}) = \frac{1}{n} \sum_{ij} \begin{cases} (1 - Y_{ij})^\alpha \log(Y_{ij}) & \text{if } \bar{Y}_{ij} = 1 \\ (1 - \bar{Y}_{ij})^\beta (Y_{ij})^\alpha \log(1 - Y_{ij}) & \text{otherwise,} \end{cases} \tag{5}$$

where $n$ is the number of objects in the image, $\alpha = 2$ and $\beta = 4$ are the focal loss weights Lin et al. (2017).

$L_{reg}$ is a gIoU loss Union (2019):

$$L_{reg}(B, \bar{B}) = \frac{1}{n} \sum_i (\text{IoU}(B_i, \bar{B}_i) - \frac{|C_i \backslash (B_i \cup \bar{B}_i)|}{|C_i|}), \tag{6}$$

where $B$ and $\bar{B}$ are the predicted and the ground truth bounding boxes of the $n$ annotated objects, $C_i$ is the enclosing convex hull of $B_i$ and $\bar{B}_i$, and $| \cdot |$ computes the area.

$L_{roi\text{-}cls}$ is a softmax classification loss on each RoI box, defined on the predicted class logits $\mathbf{c} \in \mathbb{R}^2$ and the ground truth label $\bar{c} \in \{0, 1\}$. Here we only have foreground or background classification.

$$L_{roi\text{-}cls}(\mathbf{c}, \bar{c}) = -\log \text{softmax}(\mathbf{c})_{\bar{c}} \tag{7}$$

$L_{roi\text{-}reg}$ is an L1 loss between the predicted boxes $B$ and the ground truth boxes $\bar{B}$,

$$L_{roi\text{-}reg}(B, \bar{B}) = |B - \bar{B}|. \tag{8}$$

The tracking loss is a per-element binary cross-entropy loss between the predicted association matrix $A$ and the ground truth binary matrix $\bar{A}$:

$$L_{assoc} = -\frac{1}{M} \sum_{ij} (\hat{A}_{ij} \log A_{ij} + (1 - \hat{A}_{ij}) \log (1 - A_{ij})). \tag{9}$$

The captioning loss is a softmax on each predicted word over the entire vocabulary, with a label smoothing co-efficient of 0.1 following GIT Wang et al. (2022).

$$L_{caption} = \frac{1}{L} \sum_{i=1}^{L} \text{CE}(\text{Decode}(f, \bar{y}_{1:i-1}), \bar{y}_i), \tag{10}$$

where $\bar{y}$ is the ground truth caption, $L$ is the ground-truth sentence length, and $f$ is the object feature.

## B.3 RUNTIME

We report the runtime on a 16 GB V100 GPU below for a 64-frame video in Tab. 8. Our model takes 2.36 seconds to process a 64-frame video, which means 27.1 frames per second (amortizing the global trajectory captioning time over all frames). The majority of the runtime is in the autoregressive text decoding which is an inherently sequential process.

|  | Detection on all frames | Tracking on all frames | Captioning all trajectories | Total |
|---|---|---|---|---|
| Run time (ms) | 406 | 4 | 1950 | 2360 |
| Frames per second (fps) | 157.6 | 16000 | 32.8 | 27.1 |

Table 8: Wall-clock runtime of each stage of our model. The numbers are measured on a 16GB V100 GPU for a 64-frame video.

| # | COCO | VG | SMiT | Aug COCO | VidSTG (zero-shot) $AP_M$ | VLN (zero-shot) $AP_M$ | VidSTG (finetuned) $AP_M$ | VLN (finetuned) $AP_M$ |
|---|---|---|---|---|---|---|---|---|
| 0 | | | | | - | - | 54.1 | 35.1 |
| 1 | ✓ | | | | - | - | 69.1 | 36.3 |
| 2 | | ✓ | | | 17.1 | 9.9 | 68.7 | 45.9 |
| 3 | | | ✓ | | - | - | 54.8 | 38.0 |
| 4 | | ✓ | ✓ | | 18.2 | 12.7 | 68.9 | 47.2 |
| 5 | ✓ | ✓ | | | 37.4 | 19.7 | 70.8 | 46.1 |
| 6 | ✓ | | ✓ | | 36.7 | 18.1 | 69.4 | 41.3 |
| 7 | ✓ | ✓ | ✓ | | 38.2 | 19.0 | 71.2 | **48.2** |
| 8 | ✓ | ✓ | ✓ | ✓ | **39.5** | **20.1** | **71.5** | **48.2** |

Table 9: **Zero-shot (left) and finetuning (right) evaluation of our disjoint trained models with varying datasets using image dense captioning metric $AP_M$.** We show results on VidSTG Zhang et al. (2020) and VLN Voigtlaender et al. (2023). Each row is a model pretrained on the specified datasets for zero-shot evaluation and then finetuned on the downstream datasets following Tab. 3. The results are consistent with the CHOTA metric: our models trained on joint datasets perform the best.

## C    ADDITIONAL EXPERIMENTAL ANALYSIS

### C.1    $AP_M$ EVALUATION.

mAP-METEOR is the official evaluation metric used in Visual Genome Krishna et al. (2017b) dataset for dense image object captioning. This metric evaluates predictions in each frame separately, without evaluating the tracking output.

mAP-METEOR is based on the Average Precision used in object detection Lin et al. (2014); Everingham et al. (2015), but includes a caption similarity criteria for determining true positives: *i.e.* a prediction is a true positive if the Intersection over Union (IoU) with the ground truth bounding box is above a threshold, *and* if the METEOR score Banerjee & Lavie (2005) is above another threshold. We follow the same implementation and thresholds as the Visual Genome dataset[*]. *i.e.* IoU thresholds of (0.3, 0.4, 0.5, 0.6, 0.7) and METEOR thresholds of (0.0, 0.05, 0.1, 0.15, 0.2).

In our case, some objects in the datasets Zhang et al. (2020); Voigtlaender et al. (2023) only have bounding box annotations and no caption annotations (Sec. 4.1). For these objects without caption annotations, we allow any caption prediction (and therefore ignore it) by setting its METEOR score to the maximum of 1. For brevity, we abbreviated this metric as $AP_M$. We report $AP_M$ following Tab. 3 in Tab. 9. The improvements are consistent with CHOTA.

### C.2    ABLATION OF HARD TRACKING SAMPLING.

We analyze the effect of the number of sampled frames, $m$, in hard-aggregation (Sec. 3.3) in Tab. 10. With hard-aggregation, the captioning accuracy benefits from a larger number of frames $m$, thanks to longer input-sequence length. However, this also costs more GPU memory in both training and testing. We use $m = 6$ in our ablation experiments (Tab. 2) as it achieves the best accuracy. It also follows the default number of frames used in the GIT Wang et al. (2022) video captioning model.

### C.3    USING THE UVO DATASET FOR DISJOINT PRETRAINING

For the disjoint pretraining of our model (Sec. 3.4), we used Augmented COCO as our tracking dataset. Another alternative would have been to use UVO Wang et al. (2021b), which contains real-world videos, but is relatively small at only 5000 videos.

---

[*]https://github.com/jcjohnson/densecap/blob/master/eval/eval_utils.lua

| $m$ | CHOTA($\uparrow$) | DetA($\uparrow$) | AssA($\uparrow$) | CapA($\uparrow$) | $AP_M$($\uparrow$) |
|---|---|---|---|---|---|
| 1 | 53.6 | 64.3 | **66.3** | 36.1 | 68.7 |
| 2 | 54.1 | 64.3 | 66.2 | 37.3 | 68.7 |
| 4 | 54.4 | 64.3 | 65.7 | 37.8 | 68.7 |
| 6 | **54.9** | 64.2 | 65.9 | **39.1** | 69.1 |
| 8 | 54.6 | 64.3 | 66.1 | 38.4 | **69.1** |

Table 10: **Hyper-parameter sweep for number of sampled frames, $m$, for hard-tracking**. We show results on VidSTG Zhang et al. (2020) validation. The models are based on #2 of Tab. 3 right on VidSTG. Results with hard-feature aggregation get improved with more frames and get saturated with 6 frames.

| Tracking dataset | CHOTA | DetA | AssA | CapA | $AP_M$ |
|---|---|---|---|---|---|
| *Zero-shot* | | | | | |
| UVO | 24.8 | 41.2 | 52.1 | 7.1 | 33.8 |
| Aug-COCO | **31.1** | **51.4** | **59.6** | **9.8** | **39.5** |
| *Finetuning* | | | | | |
| UVO | 50.9 | 65.2 | 53.4 | 37.9 | 70.4 |
| Aug-COCO | **56.9** | **65.8** | **70.4** | **39.7** | **71.5** |

Table 11: **Results using UVO Wang et al. (2021b) as the tracking dataset.** We show both zero-shot results (top) and finetuning results (bottom) on VidSTG datasets. For reference, we also include our results of using Aug-COCO (#8 of Tab. 3). Aug-COCO performs better in both settings, motivating our choice.

Table 11 compares Aug-COCO and UVO under the setting of Tab. 3 #8, both using a default multi-dataset sampling ratio $1 : 1 : 1 : 1$. We observe that disjoint pretraining with Aug-COCO consistently performs better than UVO in both zero-shot and finetuning scenarios, thus motivating our choice to use Aug-COCO for our experiments in the main paper.

## C.4 DETAILED CAPTIONING RESULTS

The Captioning Accuracy (CapA) component of our CHOTA metric is the average of the CIDEr, METEOR and SPICE metrics. For completeness, we report each of these captioning metrics individually in Tabs. 12 and 13, for zero-shot and full-finetuning evaluation, respectively.

## C.5 VIDSTG SPATIO-TEMPORAL GROUNDING EVALUATION

Table 7 of the main paper compared to prior methods on the spatial-video grounding task on VidSTG (where the input videos were assumed to already be temporally trimmed). In Tab. 14, we report results for spatial-, temporal- and spatio-temporal grounding by reporting the Spatial IoU (sIoU), Temporal IoU (tIoU) and Video IoU (vIoU) respectively.

The sIoU assumes the video is already temporally trimmed before evaluation, thus evaluating spatial localization. Similarly, the tIoU assumes that the video is already cropped spatially around the object of interest, and only the temporal extent of the query sentence needs to be determined, thereby evaluating temporal localization. The vIoU evaluates both spatial and temporal localization.

Our model was designed for the Dense VOC task, and not grounding, and we were able to perform grounding by selecting the bounding boxes with the highest likelihood of generating the target sentence (Sec. 3.5). As shown in Tab. 14, this approach works well for spatial-grounding, outperforming prior works in terms of the sIoU. However, as our model first generates object trajectories, without taking the input query into account, it struggles more at temporal localization. Nevertheless, it still achieves competitive results compared to prior works for both the tIoU and vIoU, although our model was not designed specifically for this task like the other methods in Tab. 14 which include explicit temporal localization modules within the network.

| # | COCO | VG | SMiT | Aug-COCO | VidSTG CapA | CIDEr | METEOR | SPICE | VLN CapA | CIDEr | METEOR | SPICE |
|---|------|----|------|----------|-------------|-------|--------|-------|----------|-------|--------|-------|
| 1 | ✓ |   |   |   | - | - | - | - | - | - | - | - |
| 2 |   | ✓ |   |   | 7.8 | 4.2 | 7.1 | 12.1 | 7.4 | 2.7 | 7.4 | 12.1 |
| 3 |   |   | ✓ |   | - | - | - | - | - | - | - | - |
| 4 |   | ✓ | ✓ |   | 8.5 | 4.4 | 7.7 | 13.4 | 8.5 | 3.1 | 8.7 | 13.8 |
| 5 | ✓ | ✓ |   |   | 8.1 | 4.0 | 7.2 | 13.0 | 7.8 | 3.1 | 7.8 | 12.4 |
| 6 | ✓ |   | ✓ |   | 4.9 | 3.3 | 7.2 | 4.2 | 7.5 | 3.7 | 9.4 | 9.6 |
| 7 | ✓ | ✓ | ✓ |   | 9.1 | 5.2 | 8.3 | **13.7** | 9.0 | 3.9 | 9.2 | 13.9 |
| 8 | ✓ | ✓ | ✓ | ✓ | **9.8** | **7.0** | **9.1** | 13.3 | **9.7** | **4.6** | **9.9** | **14.6** |

Table 12: **Detailed captioning metrics of our *zero-shot evaluation* (Tab. 3 left).** We show the individual captioning metrics CIDEr, METEOR, and SPICE for each row on both datasets.

| # | COCO | VG | SMiT | Aug-COCO | VidSTG CapA | CIDEr | METEOR | SPICE | VLN CapA | CIDEr | METEOR | SPICE |
|---|------|----|------|----------|-------------|-------|--------|-------|----------|-------|--------|-------|
| 0 |   |   |   |   | 35.8 | 43.2 | 29.0 | 35.0 | 8.7 | 3.9 | 14.8 | 7.3 |
| 1 | ✓ |   |   |   | 34.8 | 41.7 | 27.5 | 35.5 | 8.2 | 6.7 | 14.0 | 3.9 |
| 2 |   | ✓ |   |   | 39.1 | 49.4 | 30.3 | 37.6 | 16.7 | 13.9 | 20.1 | 16.1 |
| 3 |   |   | ✓ |   | 31.6 | 33.8 | 27.2 | 33.6 | 14.5 | 9.6 | 19.6 | 14.2 |
| 4 |   | ✓ | ✓ |   | 39.2 | 49.9 | 30.3 | 37.5 | **17.8** | 13.9 | 21.4 | **17.9** |
| 5 | ✓ | ✓ |   |   | 38.4 | 48.3 | 30.0 | 36.9 | 17.4 | **15.1** | 20.5 | 16.5 |
| 6 | ✓ |   | ✓ |   | 38.8 | 49.6 | 29.9 | 37.1 | 11.6 | 8.2 | 17.3 | 9.5 |
| 7 | ✓ | ✓ | ✓ |   | **40.1** | **51.5** | **30.8** | 38.1 | 17.7 | 14.6 | **21.5** | 17.1 |
| 8 | ✓ | ✓ | ✓ | ✓ | 39.7 | 51.0 | 30.6 | **37.7** | 17.7 | 14.3 | **21.5** | 17.5 |

Table 13: **Detailed captioning metrics of our *finetuning evaluation* (Tab. 3 right).** We show the individual captioning metrics CIDEr, METEOR, and SPICE for each row on both datasets.

| | Validation set sIoU | tIoU | vIoU | Test set sIoU | tIoU | vIoU |
|---|------|------|------|------|------|------|
| STGRN Zhang et al. (2020) | - | - | - | 38.0 | 48.5 | 19.8 |
| STVGBert Su et al. (2021) | - | - | - | 47.3 | - | 24.0 |
| TubeDETR Yang et al. (2022) | 56.4 | 47.2 | 28.7 | 59.0 | 48.1 | 30.4 |
| STCAT Jin et al. (2022) | - | - | - | 61.7 | 50.8 | 33.1 |
| Ours (zero-shot) | 51.8 | 40.0 | 22.0 | 54.1 | 40.2 | 22.5 |
| Ours | 58.7 | 41.8 | 25.9 | 61.9 | 41.1 | 26.3 |

Table 14: **State-of-the-art comparison of spatial-temporal grounding on VidSTG.** "-" means the numbers are not reported in the paper. Our model performs competitively at this task, although it was not actually designed for it. As our model generates object trajectories without conditioning on the input query, it struggles at temporal localization, denoted by the tIoU. The spatial localization performance, denoted by the sIoU, outperforms dedicated methods for this task.

## D    LIMITATIONS

Currently, our model produces a single caption for each trajectory, and in future work, we aim to caption potentially multiple action segments within a trajectory. Also, we repurposed existing grounding datasets for our task, as annotating a new captioning dataset can be subjective. We leave annotating a Dense VOC dataset with rigorous protocols and richer captioning as a future work.

## E    BROADER IMPACT AND POTENTIAL NEGATIVE IMPACT

Our work presents a new task and model for dense video object captioning. This task represents a general technology with a wide range of potential applications. Whilst we are unaware of all potential applications of such models, it is important to be cognizant that each application has its own merits and societal implications depending on the intentions of the individuals building and using the system. For example, we believe that the Dense VOC models can be used as part of systems to improve video search and retrieval, though they could also be used in video surveillance systems too. Additionally, we note that training datasets, especially for captioning Hendricks et al. (2018); Zhao et al. (2021), can contain biases that may render models trained on them unsuitable for deployment.

