# OpenReview forum: "Dense Video Object Captioning from Disjoint Supervision"
_ICLR.cc/2025/Conference — ICLR 2025 Spotlight_

### Official Review · Reviewer_58ev · 2024-11-02

**Soundness:** 4
**Presentation:** 3
**Contribution:** 4
**Rating:** 8
**Confidence:** 4

**Summary:**

The paper proposes a new task called Dense Video Object Captioning (Dense VOC), which involves detecting, tracking, and generating captions for objects in a video. The model aims to unify spatial and temporal localization while providing a natural language description of object trajectories. The approach combines detection, tracking, and captioning into a single end-to-end model, which is shown to be more accurate and temporally consistent compared to traditional multi-stage pipelines. The authors also propose a novel training strategy using disjoint supervision from different datasets, enabling zero-shot generalization and strong performance after fine-tuning. New metrics are introduced to evaluate the combined tasks, and the model achieves state-of-the-art results on tasks like video grounding without explicit training for those tasks.

**Strengths:**

1. The proposed approach integrates detection, tracking, and captioning, which ensures more temporally consistent results and holistic video understanding compared to multi-stage pipelines.
2. The use of disjoint tasks for training allows the model to generalize well even without specific full annotations for the task, showcasing impressive zero-shot capabilities.
3. The model outperforms strong baselines and even specialized models on tasks like video grounding and multi-object tracking, demonstrating its effectiveness across various video-related tasks.
4. The introduction of new evaluation metrics (e.g., CHOTA) tailored for Dense VOC ensures a more comprehensive assessment of model performance, capturing detection, tracking, and captioning jointly.
5. The model is successfully applied to related tasks such as video grounding and person tracking, proving its versatility.

**Weaknesses:**

1.	The framework demands significant computational resources, requiring more than 32 GPUs, and the specific GPU model was not provided. While separate training offers the potential for lightweight product applications, the experiments lack consideration of this aspect.
2.	The experimental section lacks an analysis and comparison of inference efficiency across different stages of the model as well as with existing backbones.
3.	The model heavily relies on datasets that do not fully cover Dense VOC, which could limit the effectiveness of pre-training. Although the authors address this by using multiple disjoint tasks, the absence of specialized, annotated data for Dense VOC remains a challenge.
4.	The term “Dense” is somewhat confusing and seems ambiguous in the context of the task, which unifies spatial and temporal localization in video with fine-grained visual understanding through natural language. Perhaps a more precise term could better capture the essence of this process.
5.	The caption tracking process is event-oriented. Is there an analysis of different event types in this task?

If the authors can address my concerns, I am inclined to raise my rating.

**Questions:**

1. The use of disjoint datasets and diverse forms of supervision makes the training process complex and resource-intensive, posing challenges for reproduction and practical deployment. Is it possible to unify this process within a robust code infrastructure to facilitate easier incremental implementation and application?

---

> ### Author Response · Authors · 2024-11-22
> **Author Rebuttal**
>
> We thank the reviewer for the positive feedback on our model, training pipeline, comparison to the baselines, and versatility on related tasks. We address the concerns below:
>
>
> **Is it possible to unify the multi-dataset training process within a robust code infrastructure?**
>
>
> Great point! We already include the ** full code ** to reproduce our work in the supplementary, including the data-preprocessing scripts and multi-dataset training. Adding more datasets with a different annotation type is straightforward in our infrastructure (<20 lines, as exemplified in `code/input_utils.py L711`), as the different annotation types are conveniently handled by a loss mask.
>
>
> **Compute resources needed in training and inference.**
>
>
> We would like to highlight that our computational resources required by our model are not onerous, as the entire model fits into a single GPU, and we simply use more GPUs for data-parallelism to speed up training and inference.
>
>
> We trained our model using V100 GPUs with 16 GB of memory, as mentioned in Appendix B.2 (L873). We train our model with data-parallelism, using a local batch size of 1. In our experiments, we used a global batch size of 32, and therefore used 32 GPUs. Importantly, we verified that our training is still effective with the linear learning rate scaling rule [A]: Namely, we can train our model with a global batch size of 8 or 16 (therefore with 8 or 16 GPUs respectively) by linearly increasing the number of training steps and linearly decreasing the learning rate, with minimal performance change.
>
>
> For inference, we currently use GPUs with 32GB of memory, as we process a maximum of 200 frames concurrently on the VidSTG dataset. It is possible to use less GPU memory, whilst maintaining the identical result, by running the tracking part of our model in a sliding window fashion following GTR [B], but we avoided this to avoid introducing additional implementation complexity.
>
>
>
>
> **Inference efficiency across different stages.**
>
>
> Thank you for the suggestion! We report the runtime on a 16 GB V100 GPU below for a 64-frame video below.
>
>
> |               |  Detection on all frames  | Tracking on all frames   |    Captioning all trajectories   | Total |
> |-------------|------------------------------|-----------------|--------------------------------------|--------|
> | Run time (ms) | 406                   | 4               |       1950                              | 2360 |
> | Frames per second (fps)       |   157.6                         | 16000      | 32.8                                    | 27.1 |
>
>
> Our model takes 2.36 seconds to process a 64-frame video, which means 27.1 frames per second (amortizing the global trajectory captioning time over all frames). The majority of the runtime is in the autoregressive text decoding which is an inherently sequential process.
>
>
> **The absence of specialized, annotated data for Dense VOC remains a challenge.**
>
>
> We agree. As the first work in this direction, the goal of our paper is to set up benchmarks, provide a solid baseline, and attract the attention of the community. We believe pushing better performance of our Dense VOC requires curating a training dataset with complete annotations (either manually or using an automatic/ semi-automatic data pipeline). We leave this as a future work, and we are actively working on this.
>
>
> **The term “Dense” is somewhat confusing and seems ambiguous.**
>
>
> We used “dense” following the literature for “Dense object captioning” [C], as discussed in L122-132, which localizes objects spatially in an image and describes them with text, and “Dense video captioning” [D], L130-131, which localizes events temporally in a video and describes them with text. As our task localizes both in space and time, we named our task “Dense video object captioning”.
>
>
> We are happy to rename our task and title to “Spatio-Temporal Video Object Captioning” if the reviewer thinks that is clearer.
>
>
> **Is there an analysis of different event types in this task?**
>
>
> As we are taking captioning annotations from existing datasets, we take the event statistics from the original dataset papers.
>
>
> Video Localized Narratives focuses on the “story” of the actor object. The events include object actions, interaction with other objects, and attribute changes.
>
>
> VidSTG dataset focuses on human-human or human-object relation. The events highlight changes in spatial relations (e.g., move towards/away from) and action relations (e.g., watch, hold, kick, drop etc).
>
>
>
>
> **References**
>
>
> [A] Goyal et al., Accurate, Large Minibatch SGD: Training ImageNet in 1 Hour. arXiv:1706.02677, 2017
>
> [B] Zhou et al., Global Tracking Transformers, CVPR 2023
>
> [C] Johnson et al., DenseCap: Fully Convolutional Localization Networks for Dense Captioning, CVPR 2016
>
> [D] Krishna et al., Dense-captioning events in videos. ICCV 2017

---

> > ### Comment · Reviewer_58ev · 2024-11-25
> >
> > Dear Authors,
> >
> > Thank you for your thoughtful response, which has almost completely addressed my concerns.
> >
> > However, regarding the term “Dense” in Dense Object Captioning [1] and Dense Video Captioning [2] that you mentioned, it might refer to the density in terms of the number of captions and spatio-temporal detections. Their tasks are coherent and consistent within that context. In your setting, your focus seems more inclined towards a unified task and unified training, which could potentially benefit each individual task. In this case, I think neither “dense” nor “Spatio-Temporal Video Object Captioning” fully captures the essence of your excellent work. It might be better to choose a word like “unified,” with some modifiers to emphasize your specific task.
> >
> > Thank you again; I will raise my score to accept.
> >
> > [1] Johnson et al., DenseCap: Fully Convolutional Localization Networks for Dense Captioning, CVPR 2016
> >
> > [2] Krishna et al., Dense-captioning events in videos, ICCV 2017

---

### Official Review · Reviewer_zAgu · 2024-11-02

**Soundness:** 3
**Presentation:** 3
**Contribution:** 3
**Rating:** 6
**Confidence:** 4

**Summary:**

This paper proposes the new task of detecting, tracking and captioning objects in a video. The authors propose evaluation metrics for this task, a baseline method and a training strategy involving disjoint supervision.

**Strengths:**

This work tackles an important problem that was missing from the literature. The paper is well-written and easy to follow. Extensive experiments have been performed.

**Weaknesses:**

My only concern is that the end-to-end tracking algorithm (listed as a contribution) seems to be naive and not novel enough. There are other methods that perform identity association within the model (like MinVIS[1], CAROQ [2], trackformer [3]). The authors have explored some other ways of integrating temporal information/tracking in Table 2, but how about different ways of association within the model (e.g., query vector propagation in trackformer [3])?


[1] MinVIS: A Minimal Video Instance Segmentation Framework without Video-based Training, Huang et al., NeurIPS 2022.

[2] CAROQ: Context-Aware Relative Object Queries to Unify Video Instancd and Panoptic Segmentation, Choudhuri et al., CVPR 2023.

[3] TrackFormer: Multi-Object Tracking with Transformers, Meinhardt et al., CVPR 2022

**Questions:**

Please see Weaknesses.

---

> ### Author Response · Authors · 2024-11-22
> **Author Rebuttal**
>
> We thank the reviewer for the positive comments on our task and our extensive experiments. We address the concern below:
>
>
> **How does the model compare to query-propagation-based models?**
>
>
> That’s a great question! We explored query-based detectors and trackers during our development. However, despite our best efforts, we did not get close performance as our detection-and-association model. We identify two main reasons:
>
> - The Hungarian matching used in DETR-style architectures is significantly more expensive for captioning than “standard” tasks like detection and segmentation. This is because captioning outputs are conditioned on the ground truth during training due to teacher forcing. Therefore, to compute the captioning cost for one example, we need to forward the text decoder with every pair of ground-truth and predicted objects (typically $Q = 100$ queries x $N = 50$ ground-truth objects per image in Visual Genome). This means $Q \times N = 5000$ forward passes for just a single training step. “Standard” tasks require just $Q$ forward passes, and typically use a much lighter linear head as well, so it is not a bottleneck there. Due to this compute limitation, we did not manage to train a reasonable DETR-based model on the Visual Genome dataset for image-object captioning.
> - Performing detection and tracking jointly makes it more difficult to train individually for these tasks, as we do with our disjoint, multi-dataset pretraining (Section 3.4).
>
>
> As a result, it is difficult for query-based models to take advantage of the disjoint multi-dataset training like us.
>
>
> In addition, we compared to a concurrent query-based model OW-VisCap [A] (which could not leverage disjoint pre-training for the reason mentioned above) in Table 5, and showed an improvement of 3.9 CHOTA.
>
>
>
>
> **References:**
>
>
> [A] Choudhuri et al., OW-VISCap: Open-World Video Instance Segmentation and Captioning, NeurIPS 2024.

---

> > ### Comment · Reviewer_zAgu · 2024-11-26
> >
> > Thanks for the details! My concerns have been addressed and so I will keep my positive score.

---

### Official Review · Reviewer_o3Kj · 2024-11-04

**Soundness:** 4
**Presentation:** 3
**Contribution:** 3
**Rating:** 8
**Confidence:** 4

**Summary:**

This work introduces a new task and model for dense video object captioning that unifies spatial and temporal localization by detecting, tracking, and captioning object trajectories in videos. The proposed end-to-end model combines visual understanding and language generation, outperforming multi-stage pipelines in accuracy and temporal coherence. Leveraging a novel training strategy that uses disjoint, large-scale datasets for pretraining, the model achieves impressive zero-shot capability and serves as a robust base for fine-tuning. Additionally, new metrics for task assessment and successful adaptation of video grounding datasets further demonstrate the model's effectiveness in video spatial grounding tasks.

**Strengths:**

This paper introduces a novel task—dense video object captioning with unified spatial and temporal localization—bridging video understanding and natural language description in a unique way.

The end-to-end model presented is robust, outperforming multi-stage pipelines by integrating detection, tracking, and captioning into a cohesive approach that achieves notable zero-shot capabilities.

The authors clearly outline the model architecture, training strategy, and metrics, making complex components understandable and well-motivated.

By developing metrics and reusing datasets for this new task, the work contributes valuable tools and benchmarks to the video understanding community, advancing video spatial grounding tasks and surpassing previous state-of-the-art results.

**Weaknesses:**

The paper primarily focuses on a few datasets, which may not fully represent the diversity of real-world scenarios. Expanding evaluations across a broader range of benchmarks could strengthen the validity of the results.

While quantitative metrics are important, including qualitative evaluations—such as human assessments of captioning quality—could enrich the understanding of model performance and highlight potential areas for improvement.

The paper does not provide an in-depth analysis of failure cases, which limits understanding of specific scenarios where the model struggles, such as occlusions, fast motion, or complex interactions between objects.

**Questions:**

Can you provide a detailed breakdown of common failure cases observed during the evaluation? Understanding these could clarify the model's limitations and areas for improvement.

How does your model perform with unseen objects? Providing results or insights on this aspect could strengthen the claims of generalizability.

Could you elaborate on the rationale behind the chosen evaluation metrics? A more comprehensive justification would help clarify their relevance and effectiveness in measuring performance.

---

> ### Author Response · Authors · 2024-11-22
> **Author Rebuttal**
>
> We thank the reviewer for their positive feedback on our task, model, and results. We address the weaknesses below.
>
>
> **Evaluation on a broader range of benchmarks.**
>
>
> We kindly remark that our paper presents results on 3 datasets (VidSTG, Video Localized Narratives, and BenSMOT) and 2 tasks (Dense VOC and grounding). We also remark that evaluation datasets on our new task are scarce, due to the cost of collecting such annotations. In this rebuttal, we additionally evaluate our disjoint-trained model on the image object captioning dataset, Visual Genome ,that we have trained on, and compare to single-task trained models and the SOTA baseline GRiT[A] (which uses an MAE backbone rather than CLIP backbone as in our model). The results suggest that our disjoint-pretraining maintains the performance.
>
>
> |         | mAP_METEOR  |
> |-----------------------|--------------------------------------------|
> |GRiT  |                 15.5                   |
> |Our single dataset model  |                 16.3                   |
> |Our joint model             |                  16.1                  |
>
>
> **Analysis of failure cases.**
>
>
> Thank you. We briefly discussed the failure cases in Appendix A. The failure cases include misrecognizing object classes in the output captioning, and wrong object boundaries (captioning an action before it happens, as our model predicts a single caption for the whole object trajectory). While our tracking is not perfect, we do not observe a bottleneck in tracking. It is also reflected in our high AssA scores in Table 3 and Table 4 (\~70 AssA on VidSTG, \~90 AssA on VLN and BenSMOT), whose numbers are higher than typical MOT datasets (\~50 AssA on MOT-17 datasets as shown in the [HOTA paper](https://arxiv.org/pdf/2009.07736)) [B].
>
>
> **How does the model perform with unseen objects?**
>
>
> Our model is open-vocabulary by its design (i.e., outputs free-form text), and our training set covers a diverse range of object classes. However, for objects that never appear in the training text vocabulary, our model will not have the capability to output them. Using stronger, pre-trained text decoders will mitigate this issue, as language models output texts which they have encountered previously during training.
>
>
> **Rationale behind the chosen evaluation metrics.**
>
>
> We choose an evaluation metric that measures the three primary components of our DenseVOC task: namely detecting objects in a video, tracking them throughout the video, and captioning these objects.
>
> The first two, detecting and tracking objects, are typically measured in the Multi-Object Tracking (MOT) community with HOTA [B]. We therefore adopt HOTA, which consists of detection and association terms (Detection Accuracy (DetA) and Association Accuracy (AssA) respectively), and extend it with a captioning term, which we name Captioning Accuracy (CapA).
>
>
> We decided to extend HOTA, as it is already the standard in the MOT community, and therefore facilitates fairer comparison with these models, and makes use of the lessons learned by the MOT community in evaluating trackers.
>
>
> Finally, to evaluate Captioning Accuracy, we use an average of three common captioning metrics, namely CIDEr, METEOR and SPICE. We adopt these three metrics as they are common in the community, and known to be complementary to each other. Specifically, CIDEr is common for image captioning (it is the default metric for the COCO dataset), METEOR is the main metric for object captioning (ie the Visual Genome dataset [C]). SPICE [D] is known to be complementary to CIDEr and METEOR as it highlights semantics [E].
>
>
> We will use other captioning metrics if the reviewer has other alternate suggestions. Note that we report individual values of CIDEr, METEOR and SPICE in Tables 11 and 12 of the appendix.
>
>
>
>
> **References:**
>
>
> [A] We et al., GRiT: A Generative Region-to-text Transformer for Object Understanding. ECCV 2024.
>
> [B] Luiten et al., HOTA: A Higher Order Metric for Evaluating Multi-Object Tracking. IJCV 2020.
>
> [C] Krishna et al., Visual genome: Connecting language and vision using crowdsourced dense image annotations, IJCV 2017
>
> [D] Anderson et al., SPICE: Semantic Propositional Image Caption Evaluation, ECCV 2016
>
> [E] Kilickaya et al., Re-evaluating Automatic Metrics for Image Captioning, ACL 2017

---

> > ### Comment · Reviewer_o3Kj · 2024-11-25
> >
> > Thank you for addressing my concerns and resolving my doubts. The author has adequately addressed most of my queries, and I am now inclined to recommend acceptance for this work, hence considering increasing my evaluation score.

---

### Official Review · Reviewer_szRA · 2024-11-04

**Soundness:** 4
**Presentation:** 3
**Contribution:** 3
**Rating:** 8
**Confidence:** 4

**Summary:**

This paper introduces a novel task towards the field of video understanding—dense video object captioning—with the aim of advancing the comprehension of temporal and spatial information within videos. In the absence of training data for this task, the authors innovatively proposes an end-to-end approach that integrates various datasets to train different modules. The superiority of the proposed methodology is demonstrated through extensive zero-shot and full fine-tune experiments.

**Strengths:**

1. **Clear and Precise Definition of a New Benchmark**: The paper introduces a novel task aimed at achieving more comprehensive video understanding, thereby setting higher standards for a single model's capability to interpret videos. It also establishes well-defined evaluation metrics that are appropriate for this new benchmark.

2. **Design of a Concise and Effective Training Framework**: In response to the proposed task, the paper presents a streamlined, end-to-end trainable framework that effectively integrates multiple tasks such as detection, tracking, and description generation. The authors ensure seamless end-to-end training by efficiently constructing ground truth for the tracking task.

3. **Innovative Data Organization Method**: To Address the lack of a dedicated dataset for the proposed task, the authors decompose the task and combine or generate data from various sources to train different modules effectively.

4.  **Comprehensive Experimental Evaluation**: The authors conducted extensive experiments, demonstrating the advantages of their approach over simply concatenating state-of-the-art solutions for individual tasks. They highlight the effectiveness of combining multiple datasets, the superior generalization ability of the trained model, and its outstanding performance on relevant datasets. The open source of the code further strengthens the credibility of these findings.

**Weaknesses:**

1. **Clarification of Task Significance**: Although the task imposes higher demands on deep models for video understanding, the paper does not clearly articulate the associated benefits. It would be advantageous to illustrate the task's relevance in more challenging or representative application scenarios, such as sports commentary or intelligent animal monitoring. This would better underscore its significance and research value.

2. **Examination of Method Generalizability**: The authors successfully modularize the training process by breaking down the overall task into distinct sub-tasks and aligning them with suitable training datasets, as demonstrated by the experimental results. However, a more in-depth discussion on the domain differences between datasets and the rationale for task decomposition could provide a stronger basis for the broader applicability of this approach. Such an analysis would significantly enhance the paper's contribution to the research community.

3. **Correction of Typos and Formatting Inconsistencies**: The paper contains several typos and formatting inconsistencies that require attention. For example, in Section 3.5, the symbols in Equation (1) do not correspond with the preceding descriptive content. Additionally, there are inconsistencies in citation formats in the paper, for example, the format of citations in Table 2.

**Questions:**

Beyond the questions in the weaknesses, I have an additional curiosity regarding the approach. In the Spoken Moments in Time (SMiT) dataset, captions are provided at the video level. The approach of using entire frames as detection proposal and generating captions contrasts significantly with the method of using detectors on the Visual Genome dataset to identify objects and then generating captions, resulting in a noticeable domain discrepancy. I am uncertain whether this discrepancy should be addressed, as the examples provided suggest that the generated results for single targets also include interaction information with other objects, such as "A brown dog chasing an adult on the grass." Given that the input to the text decoder is supposed to be a single-track feature and the model does not explicitly model interactions between targets, could this intriguing phenomenon be attributed to the way the SMiT dataset is utilized in training?

---

> ### Comment · Reviewer_szRA · 2024-11-22
> **Request for Author Response to the Weaknesses And Questions**
>
> As the discussion phase for ICLR 2025 is nearing its conclusion, I wanted to bring to your attention that I have posed some questions and comments regarding your submission. These points are crucial for a comprehensive evaluation of your work.
>
> Please note that if I do not receive a response soon, I may need to reconsider the quality of this submission, which could potentially impact the score I assign to your submission.

---

> > ### Author Response · Authors · 2024-11-22
> > **Rebuttal coming in the next 24 hours**
> >
> > Dear Reviewer szRA,
> >
> > We sincerely appreciate your passion and responsibility in thoroughly evaluating our submission. Your insightful questions and comments are invaluable to us. We are currently preparing a comprehensive response to all reviewers with additional experiments, and will submit it within the next 24 hours.

---

> ### Author Response · Authors · 2024-11-22
> **Author Rebuttal**
>
> We thank the reviewer for the positive feedback on our benchmark, training framework, and experiments. We reply to each concern below:
>
>
> **It would be advantageous to illustrate the task's relevance in more challenging or representative application scenarios**
>
>
> Thank you for this suggestion. We have updated the introduction to mention that models trained for DenseVOC can be used for diverse real-world applications including sports analysis and commentary, animal monitoring, security and behaviour analysis among others.
>
>
> **Rationale for task decomposition, and in-depth discussion on the domain differences between datasets**
>
>
> Great point! In our paper, we chose the task decomposition and datasets based on availability of existing annotations: we decompose our task into video captioning (SMiT, 500k videos), image object captioning (VisualGenome, 80K images), and tracking (COCO, 108K images), as large-scale datasets are available for these tasks. We do realize the domain differences in these datasets, for example, the object definitions in VisualGenome are different from COCO (discussed in L454 - L458 of the original paper), and we mitigate this by ignoring the losses from datasets that have incompatible annotations. The results in Table 3 also confirms that our mixed training yields the best results in both zero-shot and finetuning evaluation, suggesting that the domain differences in our training datasets are migrated by the large-scale training.
>
>
> **Typos**
>
>
> Thank you for pointing out the typos! We have corrected them and uploaded a new version of our paper.
>
>
> **How does the model handle interactions between targets? Is it from the entire-frame training in SMiT?**
>
>
> Good question! Our answer is that it can be from using either SMiT or Visual Genome during pretraining. It is common in Visual Genome that an object caption mentions another object, for example, “a woman riding a horse” as exemplified in Figure 1 of the [GRiT paper](https://arxiv.org/pdf/2212.00280). While our model does not explicitly model the object interaction, the region feature operator RoIAlign may have sufficient receptive fields to capture information in the context. We also agree that training on SMiT amplifies this effect, and the annotations in SMiT contain more descriptions of relations that may shift the output caption distribution towards including relations.

---

> > ### Comment · Reviewer_szRA · 2024-11-22
> > **Official Comment by Reviewer szRA**
> >
> > I am satisfied with this response and I am willing to increase the score. Additionally, I noticed that the authors mentioned in other responses that they are working on a dataset for this task. I hope they can consider more scenarios where this task can be particularly effective, similar to the ones listed in the first question.

---

### Meta-Review · Area_Chair_Hs7R · 2024-12-18

**Metareview:**

Paper proposes a novel task of detecting, tracking and captioning objects in video. It has received uniformly positive ratings from the reviewers with: 3 x accept, good paper and 1 x marginally above the acceptance threshold. Reviewers agree that the task benchmark  is useful. Main concerns were with analysis of results and failure cases and somewhat incremental algorithm design. However, given that the paper is proposing a new task and is not purely algorithmic, the last concern in the opinion of AC is somewhat mitigated. Authors have adequately addressed the reviewer concerns and reviewers are satisfied and convinced by the responses (see below).

AC agrees with consensus reached by reviewers and is recommending Acceptance. Authors are encouraged to incorporate their responses to the reviewers into the main paper.

**Additional Comments On Reviewer Discussion:**

Authors have adequately addressed the reviewer concerns and reviewers are satisfied with responses. Reviewer [szRA] explicitly states that he/she is "satisfied with this response" and is "willing to increase the score". Reviewer [o3Kj] thanks the authors for "addressing my concerns and resolving my doubts" and similarly suggests that he/she is "considering increasing my evaluation score". Reviewer [zAgu] is also positive, thanking authors "for addressing my concerns and resolving my doubts". The last reviewer, [58ev] did not, unfortunately, engage in the discussion. The generally positive post-rebuttal sentiment from the reviewers has lead to the recommendation above.

---

### Decision · Program_Chairs · 2025-01-22

Accept (Spotlight)